# Bone Growth Induction in Mucopolysaccharidosis IVA Mouse

**DOI:** 10.3390/ijms24129890

**Published:** 2023-06-08

**Authors:** Estera Rintz, Angélica María Herreño-Pachón, Betul Celik, Fnu Nidhi, Shaukat Khan, Eliana Benincore-Flórez, Shunji Tomatsu

**Affiliations:** 1Nemours Children’s Health, Wilmington, DE 19803, USA; estera.rintz@ug.edu.pl (E.R.); angelicamaria.herrenopachon1@nemours.org (A.M.H.-P.); betul.celik@nemours.org (B.C.); fnu.nidhi@nemours.org (F.N.); shaukat.khan@nemours.org (S.K.); elianapatricia.benincoreflorez@nemours.org (E.B.-F.); 2Department of Molecular Biology, Faculty of Biology, University of Gdansk, 80-308 Gdansk, Poland; 3Faculty of Arts and Sciences, University of Delaware, Newark, DE 19716, USA; 4Department of Pediatrics, Thomas Jefferson University, Philadelphia, PA 19144, USA

**Keywords:** C-type natriuretic peptide, Morquio A syndrome, ossification, AAV gene therapy

## Abstract

Mucopolysaccharidosis IVA (MPS IVA; Morquio A syndrome) is caused by a deficiency of the N-acetylgalactosamine-6-sulfate-sulfatase (GALNS) enzyme, leading to the accumulation of glycosaminoglycans (GAG), keratan sulfate (KS) and chondroitin-6-sulfate (C6S), mainly in cartilage and bone. This lysosomal storage disorder (LSD) is characterized by severe systemic skeletal dysplasia. To this date, none of the treatment options for the MPS IVA patients correct bone pathology. Enzyme replacement therapy with elosulfase alpha provides a limited impact on bone growth and skeletal lesions in MPS IVA patients. To improve bone pathology, we propose a novel gene therapy with a small peptide as a growth-promoting agent for MPS IVA. A small molecule in this peptide family has been found to exert biological actions over the cardiovascular system. This work shows that an AAV vector expressing a C-type natriuretic (CNP) peptide induces bone growth in the MPS IVA mouse model. Histopathological analysis showed the induction of chondrocyte proliferation. CNP peptide also changed the pattern of GAG levels in bone and liver. These results suggest the potential for CNP peptide to be used as a treatment in MPS IVA patients.

## 1. Introduction

Genetic skeletal dysplasia conditions account for 5% of all birth defects, and only some of those diseases have bone-targeted treatment [1]. Morquio A syndrome (mucopolysaccharidosis type IV A; MPS IVA) is one of the skeletal dysplasia caused by the accumulation of glycosaminoglycans (GAGs), keratan sulfate (KS) and chondroitin-6-sulfate (C6S). These GAGs accumulated in the growth plate of patients’ bones play a critical role in growth impairment [2,3]. MPS IVA is an inherited autosomal recessive disorder caused by a deficiency of the N-acetylgalactosamine-6-sulfate-sulfatase (GALNS) enzyme (responsible for the degradation of KS and C6S). Symptoms of the disease include systemic skeletal dysplasia, marked short stature, hypoplasia of the odontoid process, tracheal obstruction, pectus carinatum (restrictive lung), kyphoscoliosis, genu valgum, and laxity of joints. Most patients eventually are wheelchair bounded from their teenage years and do not survive the third decade of life [4,5]. Effects of severe dysplasia on the quality of life include respiratory failure, instability of the cervical region causing spinal injury [4,6], and high risk of anesthetic procedures due to the narrowed airway [7]. Patients undergo several surgical interventions throughout their life due to ongoing skeletal dysplasia [8].

One of the current treatments resolves bone dysplasia in MPS IVA patients. Available therapies for the patients include enzyme replacement therapy (ERT) and hematopoietic stem cell transplantation (HSCT), in companion with supportive treatment and surgical interventions. ERT approach has several limitations as a treatment for MPS IVA: (1) no impact on bone lesions due to its avascular cartilage region, thus not correcting bone lesions in patients [9], (2) an expensive treatment (costs around $578,000 per year for 25 kg patient) [10], (3) time-consuming treatment (weekly infusions for 4–6 h) [9,11], (4) no improvement of bone growth even if the treatment was started within two months old [9,12], and (5) production of antibodies against the enzyme that leads to immune response [13,14]. Moreover, the administration of ERT to the MPS IVA mouse model also did not improve bone pathology, even though it was tested in newborn mice [15]. HSCT as a treatment may be more effective for MPS IVA patients (benefits on heart, bone mineral density, and joint laxity [16]) but still have limitations; (1) risk of mortality and morbidity (risk of graft versus host disease) (2) difficulty in obtaining a donor for every patient, (3) complications after transplantation (4) limited impact to bone growth [10,16,17]. Both ERT and HSCT therapies are based on the cross-correction mechanism of enzyme secretion into the bloodstream. The mechanism of this phenomenon is based on the ability of lysosomal enzyme-expressing cells to correct other enzyme-deficient cells through the mannose-6-phosphate receptor pathway [18]. Although the enzyme is secreted and distributed throughout the body with the vascular system, penetrating the bone cartilage remains an unmet challenge [17]. There is a high demand for the bone-penetrating agent to treat avascular bone lesions in MPS IVA patients and induce bone ossification. A small molecule, C-type natriuretic peptide (CNP), induced bone growth in animal and human disease models. The CNP peptide activates bone growth through the natriuretic peptide receptor B (NPR-B) on both proliferative and pre-hypertrophic chondrocytes [19]. Stimulation of the intracellular molecule cGMP production by CNP activates multiple pathways, resulting in bone growth (in detail in [20]). Studies suggested that CNP has therapeutic potential for skeletal dysplasia, such as achondroplasia, a genetic disorder characterized by short stature and abnormal bone development [21]. However, the short half-life of natural CNP and the need for repeated injections of modified CNP currently limit the efficacy and convenience of CNP-based therapies [22]. As a result, ongoing research is focused on developing new CNP-based therapies with longer half-lives and improved efficacy. TransCon CNP has a longer half-life and requires only weekly injections, but repeated injections are still necessary to keep the treatment effective [23]. Therefore, there is a need to develop more effective and convenient treatments for skeletal dysplasia that can provide sustained therapeutic benefits without repeated injections.

To address this issue, we propose a therapeutic approach for progressive skeletal dysplasia in the MPS IVA mouse model. The use of CNP peptide as a potential treatment for MPS IVA has not been previously tested for this condition. Our proposed therapy utilizes an AAV vector as a one-time treatment to express CNP peptides. This novel approach is advantageous because it eliminates the need for frequent administration. A single treatment with CNP delivered through a viral vector expression cassette could be a promising therapeutic approach for treating progressive skeletal dysplasia in MPS IVA patients.

## 2. Results

### 2.1. Growth Induction in MPS IVA Mice after CNP Peptide Gene Therapy

To establish the effects of CNP peptide gene therapy on the MPS IVA mouse model, we injected intravenously AAV8 vector expressing human NPPC gene peptide under housekeeping promoter CAG into 4-week-old mice (dose 3.5 × 10^13^ GC/kg) (Figure 1). MPS IVA mice did not have GALNS enzyme activity in both plasma and tissues (Appendix A).

Mice were measured weekly for weight and length (nose to tail and nose to anus). Additionally, we took pictures on the autopsy day to compare the mice (Figure 2a). The body weights of MPS IVA and WT mice were significantly different only in week 4 (Figure 2b). In the AAV8-NPPC group, the weight significantly differed over 10 weeks starting at week 7 (Figure 2b). During the experiment, mice grew in both length parameters from week 5 (Figure 2b,c). A significant difference was observed between treated (AAV8-NPPC) and untreated MPS IVA or WT mice.

Additionally, in the final week of the experiment, we performed gait analysis in the MPS IVA mouse model to see the progression of the disease [24,25,26]. We did not observe changes in gait patterns even though CNP-treated mice had higher bony lengths (Appendix A).

### 2.2. Induction of Chondrocyte Proliferation by the Delivery of AAV8-NPPC Vector

Knee joint lesions were analyzed by toluidine blue staining 12 weeks after injection of the AAV8-NPPC vector into MPS IVA mice. Untreated MPS IVA mice showed GAG storage vacuoles in the growth plate of the tibia (Figure 3c), articular disc, and meniscus (Figure 3b) region. The growth plate region was also disorganized with vacuolated chondrocytes (Figure 3a,c). After treatment of MPS IVA mice with the AAV8-NPPC vector, the number of chondrocytes increased (Figure 3i) together with a more organized structure (Figure 3i). As previously described [2], we measured the volume of cells to assess the vacuolization of chondrocytes in the tibia growth plate region. The size of the chondrocytes decreased after the treatment of the CNP peptide (Figure 3j). Furthermore, we evaluated the improvement of vacuoles and disorganized column structures in knee joints of MPS IVA mice using pathological scores, which showed a tendency of improvement in MPS IVA mice treated with the AAV vector compared to untreated mice. The pathological score of MPS IVA mice treated with AAV8-NPPC was significantly lower for vacuolization of chondrocytes in the articular cartilage area compared to those treated with MPSIVA, but not as low as the WT level (as shown in Table 1). Regarding other parameters, we observed a decrease, but it was not statistically significant.

### 2.3. AAV Vector Biodistribution and CNP Expression

To establish if the one-time administration of vector expressing CNP affects CNP expression during the study, we measured the plasma level of human NT-proCNP as a marker of CNP expression. The expression level of NT-proCNP increased to the level of 1860.8 ± 584.6 pmol/L just four weeks after the injection (Figure 4a). Over time (weeks 12 and 16), the levels of the NT-proCNP increased, but it was not statistically different from the 8-week-old measurement. In the case of WT and untreated MPS IVA mice, human NT-proCNP was not detected. In addition, we measured the biodistribution of the vector in the liver and bone (humerus). The vector was detected in both liver (Figure 4b) and bone (Figure 4c).

We performed immunohistochemistry staining to investigate the presence of CNP peptide in the liver and bone growth plate. However, we did not detect CNP in the bone growth plate samples, possibly because the peptide was secreted by the cells. In the liver samples, immunohistochemistry staining showed no significant difference in the expression levels between the mouse groups. These results could be associated with antibodies that cross-react with the murine NPPC gene.

In addition, we performed a Western blot to quantify the expression of the CNP protein in the liver, as its distribution level was over three-fold higher than in WT mice. We detected CNP expression in all groups, including MPS IVA, WT, and AAV8-NPPC (shown in Appendix A). The antibody used detected both human and mouse CNP; however, we observed the highest expression in AAV8-NPPC-treated mice due to the expression of both human and endogenous mouse CNP.

### 2.4. GAG Levels in the Blood and Tissues

At the autopsy, we measured mono-sulfated KS, the major KS component, in both plasma and tissues (liver, heart, lung, and bone). The levels of mono-sulfated KS are shown in Figure 5. KS was accumulated in plasma and all tissues except the heart (Figure 5c) of the MPS IVA mouse model. After treatment with CNP expressing vector, plasma KS level statistically decreased compared to untreated mice (Figure 5a). The level of KS is lower in the bone of CNP-treated mice compared to untreated MPS IVA (Figure 5c). Treatment with vector did not influence the levels of KS in the heart and lung (Figure 5c,d).

Additionally, we measured the levels of other GAG in the tissues and plasma; Di-sulfated KS (Di S KS), Chondroitin disaccharide (Di 4S), O-sulfated heparan sulfate, and low sulfated heparan sulfate. We did not observe significant differences between the groups in the di-sulfated KS, O-sulfated heparan sulfate, and low-sulfated heparan sulfate. The chondroitin sulfates were accumulated in the liver of the MPS IVA mouse. Treatment with CNP expressing vector did not affect the GAG levels in the liver, but in the case of the lung after the treatment, we observed higher levels compared to WT and untreated MPS IVA mice.

### 2.5. Micro-CT

Micro–Computed Tomography (micro-CT) is an important tool for analyzing cortical and trabecular bone structure in medical applications such as treatment efficacy and bone tissue engineering. Micro-CT uses X-ray attenuation data to create a 3D representation of the specimen, providing direct 3D measurements of bone morphology and a larger volume of interest than traditional histologic evaluations. Micro-CT measurements are also faster and nondestructive, allowing other assays to be performed on the same sample [27]. To determine bone morphometry, we performed micro-CT in three experimental groups with analysis of the trabecular and the cortical bone (Figure 6).

We did not see significant differences between WT and untreated MPS IVA mice, partly because of the limited number of mice (Analysis of Trabecular Bone Morphometry *n* = 5 in each group (Figure 6d–j); Cortical Bone Architecture WT; *n* = 5, MPS IVA; *n* = 5, AAV8-NPPC; *n* = 4 (Figure 6n–s). AAV8-NPPC treatment increased the trabecular volume of interest (Figure 6d) and decreased trabecular thickness (Figure 6g). In cortical bone analysis, the total cortical medullary area increased after the treatment (Figure 6q). Additionally, we measured the length of the femur used for analysis (Figure 6t). No bone length difference between untreated MPS IVA and WT mice was found. While MPS IVA mice were treated with AAV8-NPPC, femurs were longer (Figure 6t).

## 3. Discussion

The histological structure of the growth plate region in MPS IVA patients showed vacuolated cartilage cells with unorganized column structure and reduced calcification [28]. The short stature of MPS IVA patients is directly linked to the abnormal lysosomal storage within the chondrocytes, reducing their ability to properly proliferate and differentiate [29]. Our MPS IVA mouse model showed mono-sulfated KS accumulation in the bone (Figure 5) and an abnormal histological picture (Figure 3). Similarly, as in patients [30,31], the tibial growth plate showed an accumulation of GAG in vacuolated chondrocytes and abnormal column structure. The results were also parallel as in our other mouse models (MKC and MTOL, discussed in [2]); tibia growth plate chondrocytes were found to be more prominent in MPS IVA knockout mice than in WT (Figure 3). Furthermore, our analysis delved more deeply into the level of GAG accumulation in affected tissues. We were the first to show mono-sulfated KS accumulation in the bone of the mouse MPS IVA model, in addition to other tissues. The difference in the level of mono-KS in the liver and lung (the only tissues shown in [2]) was similar. In untreated MPS IVA, liver accumulation was 200 times higher than in WT, while in the lung, it was around 100 times more. These results are similar to our previous mouse models [2]. In the case of bone, we observed a 30% increase in mono-KS accumulation compared to WT mice (Figure 5).

As a result of a larger deletion in the GALNS gene, we did not detect the GALNS enzyme activity in any tissues and plasma samples of the MPS IVA mouse model (Appendix A). Nevertheless, with a limited number of mice analyzed, we did not observe abnormal bone growth and unique skeletal dysplasia in the new MPS IVA mouse model (Figure 2) and significant changes in the morphometry of mouse bone (Figure 6). However, there was a tendency in the MPS IVA vs. WT: higher VOI (Figure 6d), higher bone volume (Figure 6e), a higher percent of bone volume (Figure 6f), lower trabecular separation (Figure 6h), and higher bone area (Figure 6n). We will require a more significant number of mice to analyze the bone morphometry.

In an article comparing different bone lesions in MPS types I, IIIA, IVA, and VII based on micro-CT scans, it was found that MPS VII showed the highest bone abnormalities, including severe thickening of the periosteum of vertebrae and bones in the knee joint. This thickening resulted in a rough articular surface in the knee joint bones. Both plain radiographs and BMD measurements by micro-CT showed an increase in MPS VII compared to WT mice. The second most severe bone dysplasia with similar outcomes was observed in MPS I. In the case of MPS IIIA, no difference was observed in micro-CT scans compared to WT mice. While micro-CT scans of MPS IVA mice showed no significant abnormalities compared to WT mice, an abnormality in the calcaneus bone was detected on radiographs of nine-month-old MPS IVA mice, and the same anomaly was clinically evident in four other mice at six to seven months of age [32].

Overall, our experimental mouse model has similar abnormal characteristics to patients with MPS IVA [33,34], including no enzymatic activity, accumulation of GAGs, and histological changes in the bone of mice. To this date, the experimental mouse models have not shown severe skeletal phenotypes similar to that in human patients.

Many forms of mucopolysaccharidosis (MPS), such as types I, II, III, VI, and VII, have suitable animal models that accurately represent their phenotype. These models include genetically modified animals and spontaneous mutation that occurs in feline and canine models [35]. However, for MPS IVA, no spontaneous model has been described, and to this day, the knockout mouse models have shown a limited skeletal phenotype.

In our laboratory, we have developed several mouse models with differences in the deletion size and place of the GALNS gene [2,36]. Our experience confirms that mouse models with the knockout in the GALNS gene have pathological bone phenotypes in histological analysis, along with progressive KS accumulation, which is necessary to reflect human pathology [2,36,37]. Recent studies have shown a rat model of MPS IVA generated using CRISPR/Cas9 technology. In comparison to the current mouse models, the rat model showed skeletal alterations that more closely resemble those of human patients [38].

The application of recombinant AAV (rAAV) vectors in gene therapy to address MPS is highly valuable due to their efficient infection of various cell types, capacity to persist as episomes, and minimal risk of causing insertional mutagenesis or genotoxicity. The AAV vector transduction is a multistep mechanism with crucial steps such as cellular internalization, endo-lysosomal vesicle uptake to the cell center, capsid processing for the nuclear AAV capsid entry following uncoating and AAV replication with the expression of the transgene [39]. AAV vectors have several serotypes; to date, 12 of the primate serotypes have been described (AAV1-12) [40,41,42,43,44,45]. Each serotype differs in capsid sequences and determines its tissue tropism and antigenic properties [45,46]. This study used an AAV8 vector with liver and heart tropism [47]. As the AAV8 vector internalized to the cell, the expression of the NPPC gene started in the nucleus to produce CNP peptide outside the cell membrane. The synthesis of the CNP peptide is a process started in the nucleus, where the transcribed sequence is cleaved by the enzyme furin to be secreted outside the nucleus in the propeptide CNP form. The CNP propeptide (103 amino acids) was cleaved to biologically form active CNP and inactive N-terminal proCNP (NT-proCNP) peptides, and those forms are secreted outside the cell [48,49]. As NT-proCNP has a longer life with a significant correlation to the biologically active CNP concentration, this molecule indicates CNP expression [50]. We detected increased expression of human NT-proCNP in the circulation of mice (Figure 4a); however, we did not detect CNP expression in the bone of treated mice following IHC staining. This finding suggests that CNP peptide was expressed from the AAV cassette, attached to the NPR-B receptor on the top of the cell, and activated a cascade of molecular reaction that resulted in the growth induction in mice [more on the mechanism of bone growth induction: 20]. CNP also regulates cellular condensation and GAG synthesis during the chondrogenesis process in the in vitro culture [51,52]. CNP increases the expression of enzymes involved in chondroitin sulfate synthesis, which is necessary for cartilage GAGs. In this study, authors did not see the change in other genes responsible for the extracellular matrix (Sox9, −5, −6, nor collagen II) [51]. Similar studies also showed an increase in GAG synthesis based on the Alcian staining; in that case, it was highly dose-dependent (10^−8^ M and 10^−7^ M CNP increased GAGs, while 10^−6^ M CNP did not affect) [52]. Our results showed decreased KS accumulation in bone compared to the untreated MPS IVA (Figure 5e). There is no direct explanation for why CNP could decrease KS in bone. Pathology slides showed a smaller size and increased number of chondrocytes after CNP treatment (Figure 3). We hypothesize that smaller chondrocytes cannot store as many KS as ballooned chondrocytes in the MPS IVA mouse model, which can indirectly explain the decreased accumulation of KS in bone tissue after CNP treatment. However, we have a limited number of animals in the pathology assessment. The overexpression of CNP previously confirmed its role in the bone ossification process by inducing bone growth in transgenic mice [53]. This research showed a decrease in bone mineral density [53]. We observed a decrease after the treatment, but it was not statistically significant (Figure 6). Natural CNP has a short lifetime of 2–3 min with a rapid degradation process that results in the ineffective long-term treatment of achondroplasia. One of the first studies to use CNP as a potential treatment for skeletal dysplasia was done in a mouse model of achondroplasia, the most common form of dwarfism caused by mutations in the fibroblast growth factor receptor 3 (FGFR3) gene. The study found that CNP could act as a negative regulator of FGFR3, inhibiting mitogen-activated protein kinase (MAPK). After administration of the CNP analog to the mouse model of achondroplasia, the mice were much longer than their affected littermates. The mouse growth plates showed an increase in chondrocytes, and immunohistochemistry showed that after CNP treatment, the growth plates were partially normalized, together with enlargement of the epiphysis, to the level of wild-type mice. Both the proliferative and hypertrophic zones of the growth plate showed an increase in height after CNP treatment [54]. These results are similar to our histological analysis of the MPS IVA mouse model after CNP expression (Figure 3).

As a continuum of the mouse research in achondroplasia clinical trial for the Vortosite was done. In the phase 2 clinical trial, treatment with vosoritide for one year resulted in a statistically significant increase in growth velocity compared to placebo in children with achondroplasia. The most common adverse events associated with vosoritide were injection site reactions and mild hypotension, which were generally well-tolerated. Currently, Vortosite is accepted as a treatment for achondroplasia patients aged ≥ two years whose epiphyses are not closed. Nevertheless, patients must receive subcutaneous injections daily to sustain CNP activity [21]. What may result in long-term side effects is not shown. As a result, another modified CNP therapeutic, TransCon, is tested in clinical trials for longer-half life (weekly injection) and efficacy [23]. Repeated injections are still necessary to keep the treatment effective; therefore, there is a need to develop more effective and convenient treatments for skeletal dysplasia that can provide sustained therapeutic benefits without the need for repeated injections.

Our study tested the potential of CNP in the expression cassette of AAV for the first time.

While our study provides valuable insights into the potential therapeutic use of CNP for progressive skeletal dysplasia in MPS IVA, several limitations exist. Firstly, our study used a small sample size, which could affect the statistical power and significance of the conclusions drawn. Further studies with larger sample sizes may be necessary to confirm the effectiveness of CNP delivered through an AAV vector expression cassette. It should also be noted that the animal model we used did not exhibit the typical phenotype of MPS IVA patients. While none of the existing mouse models fully replicate the phenotype of human MPS IVA, our study used a model closer to growth plate pathology. Lastly, our study only evaluated the short-term effects of CNP treatment and did not assess potential toxicity or long-term outcomes. Further toxicity and long-term effects analysis will be necessary before CNP is delivered through an AAV vector expression cassette, which can be considered a viable therapeutic approach for progressive skeletal dysplasia in MPS IVA patients.

Overall, we tested for the first-time usage of CNP in an AAV cassette in the MPS IVA. The administration of AAV8 expressing CNP peptide into the MPS IVA mouse model resulted in (1) high secretion of CNP peptide into circulation, (2) induction of bone growth, (3) proliferation of chondrocytes, (4) improvement in bone pathology, and (5) changes in the levels of GAGs.

## 4. Materials and Methods

### 4.1. Experimental Design

All procedures for this study were approved by the Institutional Animal Care and Use Committee at Nemours Children’s Health (IACUC; RSP19-12482-001). Mice were housed in a 12/12 h light/dark cycle with food and water provided ad libitum. The experimental design is shown in Figure 1b, with three experimental groups (*n* = 5, unless stated otherwise): (1) MPS IVA mice treated with CNP expressing vector, (2) WT mice (WT), and (3) untreated MPS IVA mice (UT). Four-week-old male mice were injected intravenously with an AAV vector expressing CNP peptide at a dose of 3.5 × 10^13^ GC/kg in PBS (Vector Builder). After AAV gene therapy, length, and weight were measured weekly for the mice, and blood was collected biweekly for 12 constitutive weeks. At 16 weeks old, mice were euthanized in a CO_2_ chamber and perfused with 10 mL of 0.9% saline. Tissues were collected for GALNS enzyme activity assay, GAG analysis, and immunohistology staining.

### 4.2. Expression Vector

Adeno-associated virus vector used to overexpress the human native NPPC gene in our study, pAAV[Exp]-CAG > hNPPC[NM_024409.4]:WPRE, was constructed and packaged by VectorBuilder (Figure 1a). Briefly, for the recombinant AAV manufacturing, the transfer plasmid carrying the gene of interest (GOI) was co-transfected with the proprietary Rep-cap plasmid and helper plasmid encoding adenovirus genes (E4, E2A, and VA) that mediate AAV replication into HEK293T packaging cells. After a short incubation period, viral particles were harvested from cell lysate or supernatant depending on serotype and concentrated by PEG precipitation. For ultra-purified AAV (in vivo grade), viral particles were further purified and concentrated by cesium chloride (CsCl) gradient ultracentrifugation. We used a qPCR-based approach to measure AAV titer.

### 4.3. MPS IVA Mouse Model

MPS IVA knockout mice (MKC2; C57BL/6 background) were generated with a larger deletion site than the previously described murine model of MPS IVA [55]. Two pairs of sgRNAs were cleaved together to generate a large deletion (~6300 bp) between the two target sites (up- and downstream of the GALNS gene) (Appendix A). GALNS enzyme activity is not detectable in tissues and blood (Appendix A). Due to GALNS deficiency, mice accumulate storage material in tissues (liver, heart, lung, bone). KS level and consequent pathology are widely used in mouse models to evaluate the severity of the patient’s phenotype and improvement of the treatments [36,56]. Genotyping for the experimental cohorts was done by PCR on day 20. Primers are flanking each sgRNA site designed to test individual nonhomologous DNA end joining (NHEJ) activity, as well as paired together to screen for deletion mutations between the two target sites.

### 4.4. GALNS Enzyme Activity Assay

GALNS enzyme activity was determined in plasma and tissues [2]. Frozen tissues were homogenized by Bead Mill Homogenizer (OMNI International, Kennesaw, GA, USA) in 25 mmol/L Tris-HCl (pH 7.2) and 1 mmol/L phenylmethylsulphonyl fluoride. Then, homogenates were centrifuged for 30 min at 4 °C, and the supernatant was collected to a new tube and assayed for enzyme activity. Either tissue lysate or plasma (2 μL) was used for the enzymatic reaction with 22 mM 4-methylumbelliferyl-β-galactopyranoside-6-sulfate (Research Products International, Mount Prospect, IL, USA) in 0.1 M NaCl/0.1 M sodium acetate (pH 4.3) and incubated at 37 °C for 16 h. After incubation, 10 mg/mL β-galactosidase from *Aspergillus oryzae* (Sigma-Aldrich, St. Louis, MO, USA) in 0.1 M NaCl/0.1 M sodium acetate (pH 4.3) was added and incubated for additional 2 h at 37 °C. The reaction was stopped with 1 M glycine NaOH (pH 10.5) solution and read at excitation 366 nm and emission 450 nm by FLUOstar Omega plate reader (BMG LABTECH Inc., Cary, NC, USA). Activity is shown as nanomoles of 4-methylumbelliferone released per hour per microliter of plasma or milligram of protein. Protein concentration was determined by a bicinchoninic acid (BCA) protein assay kit (Thermo Fisher Scientific, Waltham, MA, USA).

### 4.5. Glycosaminoglycans Quantification

GAG assay was determined in plasma and tissues as previously described [2].

### 4.6. Gait Analysis

Gait analysis was performed as described previously [57]. Briefly, mouse feet from forelimbs were painted with orange paint, and feet from hindlimbs were painted with green paint. After that, we used paper to track animal footprints. We defined stride length as the distance between two sequential footprints of hindlimbs created by the same foot (called one stride). To stride width, we measured the distance between the left and right hindlimbs.

### 4.7. Bone Pathological Assessments

Tissue staining and analysis were performed as described by Tomatsu et al. [2]. Briefly, knee joints were collected from 16-week-old MPS IVA and WT mice to evaluate levels of storage granules by light microscopy. Tissues were fixed in 2% paraformaldehyde and 4% glutaraldehyde in PBS, post-fixed in osmium tetroxide and embedded in Spurr’s resin. Then, toluidine blue-stained 0.5-µm-thick sections were examined. To evaluate chondrocyte cell size (vacuolization) in the growth plates of the femur or tibia, approximately 300 chondrocytes in the proliferative area were measured in each mouse by Image J 1.53t software, and results were expressed as folds-change from the wild-type group. Pathological slides from knee joints of treated and untreated MPS IVA and wild-type mice were evaluated for reduced vacuolization and improved column orientation in growth plates (Table 1). The number of storage materials and the degree of disoriented columns were scored. “No storage or very slight” was 0 (−), “slight but obvious” was 1 (+), “moderate” was 2 (++), and “marked” was 3 (+++) [2]. Each pathological slide was assessed in a double-blind manner three times. We then averaged the scores in a group of mice per section of bone (growth plate, articular disc, meniscus, and ligament).

### 4.8. Trabecular and Cortical Morphometry

Trabecular and Cortical Morphometry was performed by micro-CT scans (a Bruker SkyScan 1275 scanner (Bruker, Billerica, MA, USA). Before analysis, the right femur of 16-week-old mice was fixed with 96% ethanol. For analysis, the femur was wrapped in 0.9% NaCl gauze and scanned. Quantitative analysis was performed in Bruker CTan software (v1.21.1.0). The volumes of interest (VOI) of trabecular bone were identified using the distal epiphyseal plate. The volumes of interest (VOI) of cortical bone were determined based on the distal epiphyseal plate start mark and the highest point in the proximal greater trochanter; between those two landmarks, we could obtain the total length of the bone.

### 4.9. AAV Vector Genome Biodistribution

DNA vector biodistribution was determined as previously described [2]. Briefly, DNA was purified from the liver using the Gentra Puregene kit according to the instruction manual (QIAGEN, Germantown, MD, USA). Before purification, the bone was homogenized by Bead Mill Homogenizer (OMNI International, GA, USA) and purified according to manufacturer instructions (QIAGEN, Germantown, MD, USA). Digital PCR (dPCR) analysis on genomic DNA extracted from liver and bone samples of mice was performed using specific primers and probe sequences for NPPC gene, which are as follows: forward primer, AAC GCG CGC AAA TAC AAA G, reverse primer, GGA ATT CCC ACT TTG TAC AAG AAA, and probe 5′6-FAM /TG AGC GGC C/ZEN/T GGG ATG TTA GAC CCA/3′IABkFQ. Genomic DNA was tested non-fragmented and fragmented using enzymatic digestion or an M220 Focused-ultrasonicator (Covaris, Woburn, MA, USA). The NPPC TaqMan assay (FAM-labeled) (Thermo Fisher Scientific, Waltham, MA, USA) was used for quantitative dPCR analysis of the AAV CNP expressing vector. The concentration of DNA for the AAV chip for the liver and bone samples processed was between 0.5 and 2 ng per 16-mL reaction, dependent on the DNA concentration needed to bring the AAV copies/mL into the detectable range of the instrument. The Tfrc TaqMan copy number reference assay (VIC labeled) was obtained from Thermo Fisher Scientific. 40 ng of genomic DNA per 16-mL reaction was used for Tfrc dPCR. Each reaction was loaded onto a separate QuantStudio chip (QuantStudio 3D digital PCR 20 K chip kit v2, A26316; Thermo Fisher Scientific, Waltham, MA, USA). The PCR amplification profile was conducted with an ABI GeneAmp 9700 PCR thermal cycler with dual flat blocks (Applied Biosystems, Waltham, MA, USA). Then, chips were read on the QuantStudio 3D instrument to obtain the number of wells positive for the VIC and FAM channels and the number of wells without DNA and empty wells. Data analysis and chip quality were assessed using the QuantStudio 3D analysis suite. All chips were between 25% and 75% empty wells, ensuring suitability for quantitation. Copies/mL for both Tfrc and AAV were determined and normalized using the dilution factor for AAV sample input. Using Tfrc results as a reference for two copies, the number of copies of AAV per mouse genome was calculated.

### 4.10. ELISA for NT-proCNP

The detection of NT-proCNP in mice was done according to manufacturer instructions (Biomedica Medizinprodukte, Wien, Austria). Sample preparation included blood collected from mice and then centrifuged 8000× *g* at 4 °C for 10 min to collect the plasma sample in another tube and stored at −20 °C until further analysis. Plasma used for NT-proCNP detection was diluted 1:100 in the assay buffer provided in the ELISA kit.

### 4.11. Western Blot

Protein lysate was isolated from both the liver and arm bone, as described in Section 4.4. After lysis and determination of protein concentration, the protein extracts were separated by electrophoresis. However, we did not obtain a sufficient protein concentration in the arm tissue for further analysis. Therefore, only the liver tissue was analyzed for protein expression. After electrophoresis, the proteins were transferred onto a nitrocellulose membrane. The membrane was blocked with 5% nonfat dry milk in a TBST buffer and then incubated with primary antibodies overnight at 4 °C (anti-CNP antibody from MyBioSource, San Diego, CA, USA, and beta-actin from Sigma-Aldrich, USA). The membrane was subsequently incubated with secondary antibodies at room temperature for 1 h, treated with a solution of substrates for HRP detection, and read with a C-DiGiT Blot Scanner Licor Machine. As CNP and beta-actin have similar protein weights, the membrane was stripped after performing analysis for the CNP protein, and beta-actin analysis was performed starting from the blocking step. The intensities of bands were analyzed using the QuantityOne 29.0 software.

### 4.12. Statistical Analysis

The normal distribution was found using the Kolmogorov-Smirnov test (K-S) and homogeneity of variance with the Leven test. Depending on the results, analysis of variance (two-way ANOVA) with Tukey’s post-hoc test was performed if the distribution was normal. When the assumptions of the normality of the distribution and the homogeneity of variance were unmet, the nonparametric Kruskall-Wallis test, followed by Dunnett’s test, was performed. Statistical analyses were performed using GraphPad Prism 9 software. Statistical significance was set at *p* > 0.05. Error bars represent SEM, as indicated in legends.

## Figures and Tables

**Figure 1 ijms-24-09890-f001:**
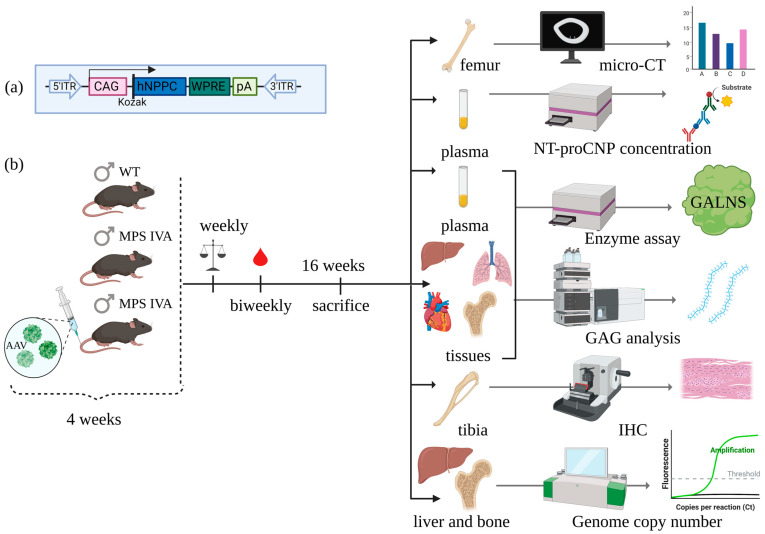
Experimental design. (**a**) A viral vector expressing cassette. (**b**) Experiments started at four weeks old. We have three experimental groups: untreated MPS IVA, wild-type control (WT), and treated MPS IVA with the AAV8 vector expressing CNP peptide (AAV8-NPPC). During the experiments, mice were measured weekly for body weight, nose-tail length, and nose-anus length, with biweekly blood collection. Mice were euthanized at the age of 16 weeks. After that, the analysis was made. This figure was prepared using BioRender.com.

**Figure 2 ijms-24-09890-f002:**
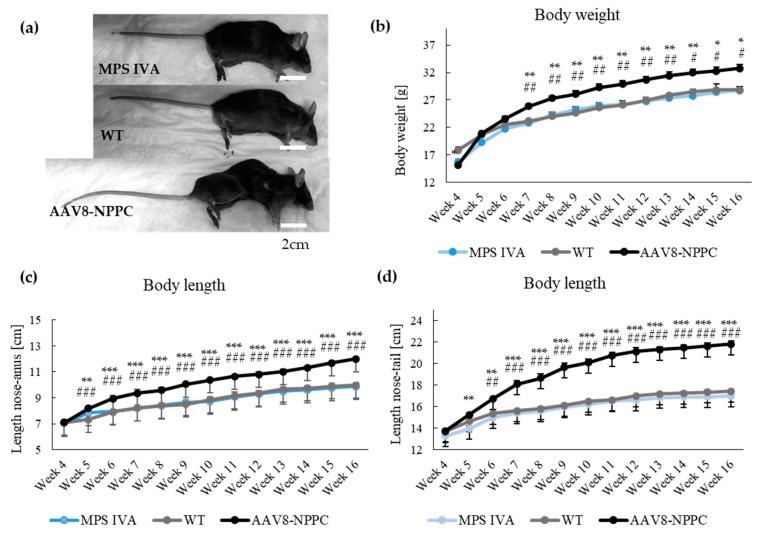
Establishment of CNP peptide growth induction in MPS IVA mouse model. (**a**) Picture of mice at week 16. (**b**) Body weight of mice during the experiment. (**c**) Body length measurement from nose to anus of mice during the experiment. (**d**) Body length measurement from nose to tail of mice during the experiment. Results are shown as mean values ± SEM (*n* = 5). The following statistical symbols were used to denote as follows. AAV8-CNP group vs. WT group, *** *p* ≤ 0.001, ** *p* ≤ 0.01, * *p* ≤ 0.05; AAV8-CNP group vs. untreated MPS IVA group, ### *p* ≤ 0.001, ## *p* ≤ 0.01, # *p* ≤ 0.05.

**Figure 3 ijms-24-09890-f003:**
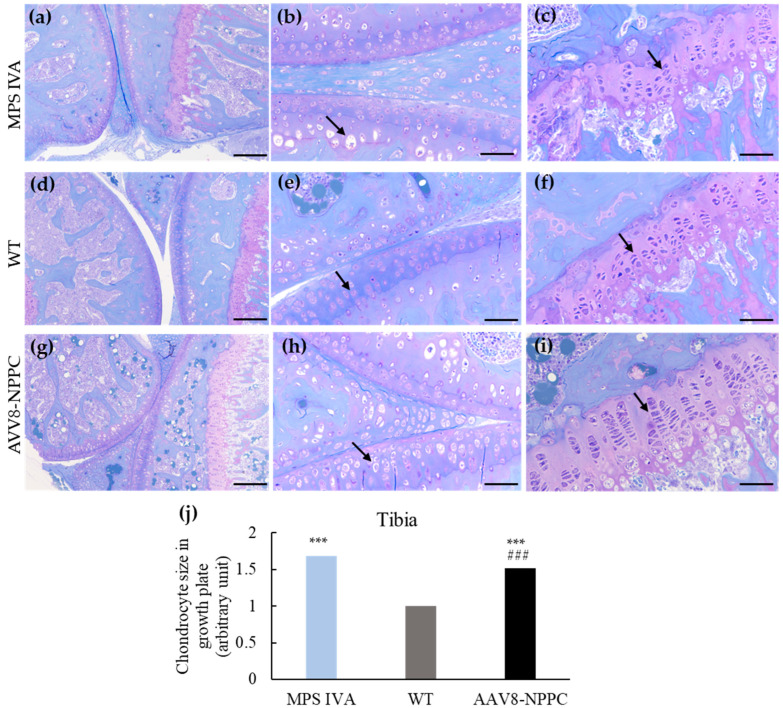
Correction of bone pathology in MPS IVA mice treated with AAV8-NPPC vector assessed by toluidine blue staining analysis using light microscopy. Knee joint and articular cartilage of untreated MPS IVA (**a**), WT (**d**), and MPS IVA mouse treated with AAV8-NPPC vector (**g**) (10× magnification and scale 200 µm). Articular disc in the knee joint of MPS IVA (**b**), WT (**e**), and MPS IVA mouse treated with AAV8-NPPC vector (**h**) (40× magnification and 50 µm). Growth plate region of MPS IVA (**c**), WT (**f**), and MPS IVA mouse treated with AAV8-NPPC vector (**i**) (40× magnification and 50 µm). Chondrocyte cell size in growth plates of the tibia (**j**). Data expressed fold change from the WT group (*n* = 3). Arrows indicate chondrocytes. The following statistical symbols were used to denote as follows. AAV8-CNP group vs. WT group, *** *p* ≤ 0.001; AAV8-CNP group vs. untreated MPS IVA group, ### *p* ≤ 0.001.

**Figure 4 ijms-24-09890-f004:**
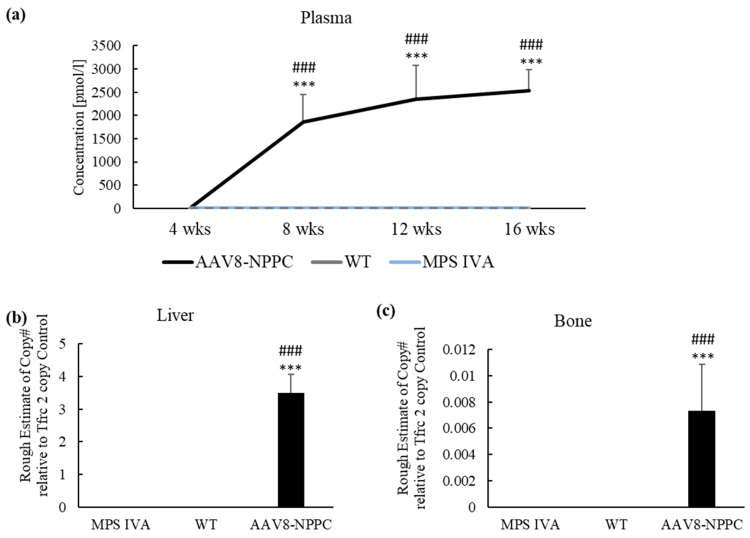
AAV vector Biodistribution and NT-proCNP expression and vector copy number at 12 weeks post-treatment. (**a**) Plasma; Marker for CNP expression, NT-proCNP, increased over the 12 weeks of treatment in the MPS IVA mouse model with significant differences to WT and untreated MPS IVA mice. (**b**) Liver and (**c**) bone; vector copy number at 12 weeks post-treatment. Results are shown as mean values ± SEM (*n* = 5). The following statistical symbols were used to denote as follows. AAV8-CNP group vs. WT group, *** *p* ≤ 0.001; AAV8-CNP group vs. untreated MPS IVA group, ### *p* ≤ 0.001.

**Figure 5 ijms-24-09890-f005:**
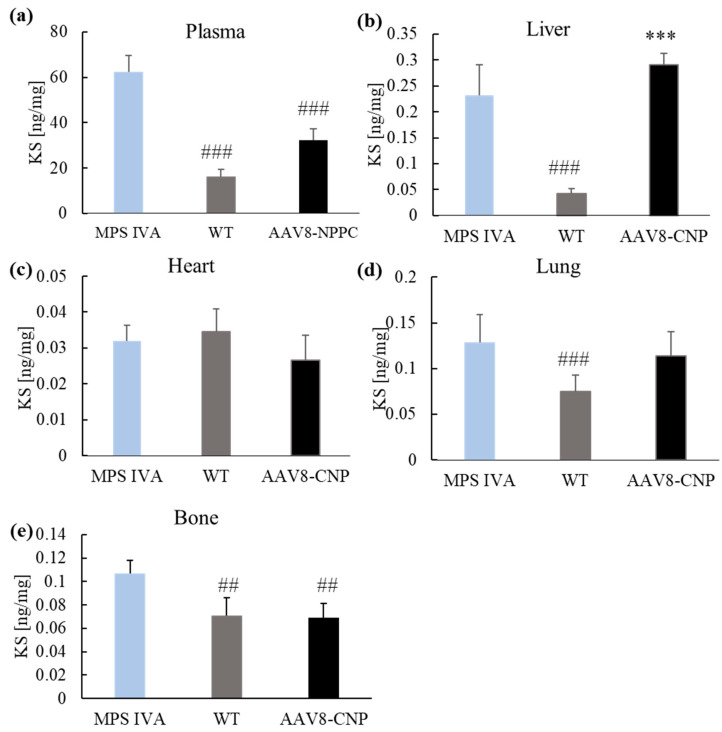
Mono-sulfated keratan sulfate levels. After 12 weeks post-treatment, the KS level was measured in plasma (**a**), liver (**b**), heart (**c**), lung (**d**), and bone (**e**). Results are shown as mean values ± SEM (*n* = 5). The following statistical symbols were used to denote as follows. AAV8-CNP group vs. WT group, *** *p* ≤ 0.001; AAV8-CNP group vs. untreated MPS IVA group, ### *p* ≤ 0.001, ## *p* ≤ 0.01.

**Figure 6 ijms-24-09890-f006:**
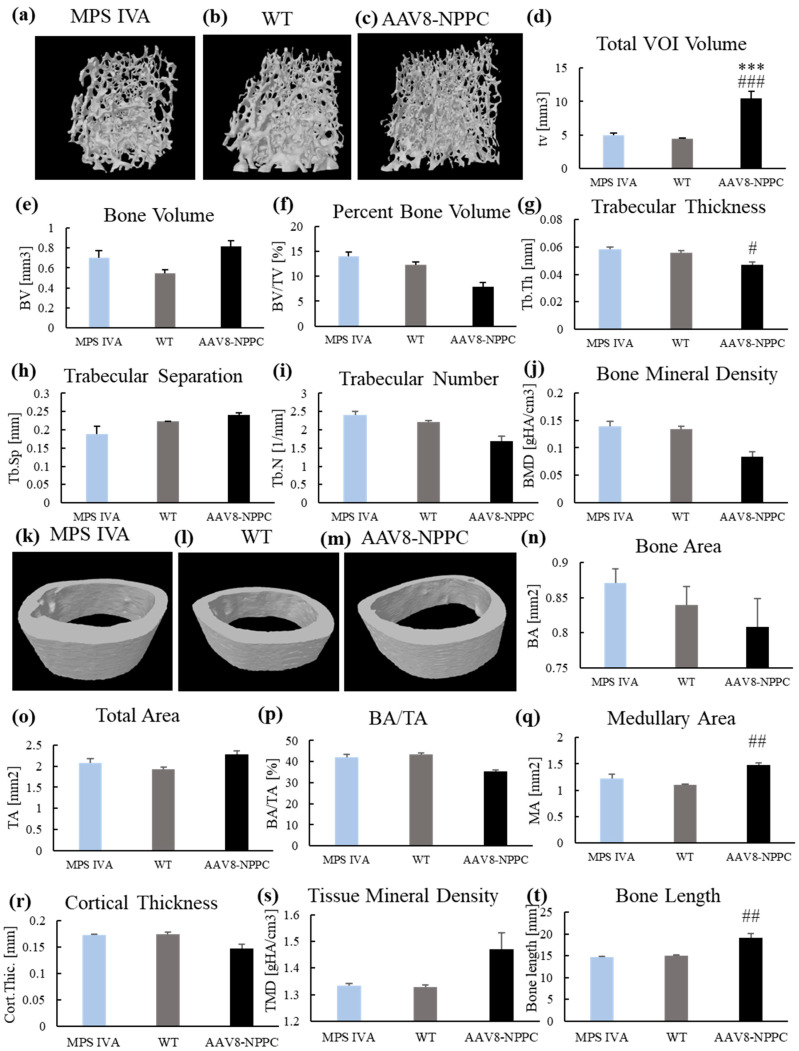
Trabecular and cortical morphometry after AAV vector treatment. At 12 weeks after treatment, micro-CT analysis of trabecular (**a**–**j**) and cortical (**k**–**t**) bone of three experimental groups. Trabecular analysis included (*n* = 5): trabecular bone model of MPS IVA (**a**), WT (**b**), and AAV8-NPPC (**c**); total trabecular volume of interest (**d**); trabecular bone volume (**e**); percent bone volume (**f**); trabecular thickness (**g**); trabecular separation (**h**); trabecular number (**i**); bone mineral density (**j**). Cortical analysis included (AAV8-NPPC; *n* = 4; MPS IVA, *n* = 5; WT; *n* = 5): cortical bone model of MPS IVA (**k**), WT (**l**), and AAV8-NPPC (**m**); Cortical bone area (**n**); Total area (**o**); Bone area/Total area (**p**); Medullary area (**q**); Cortical thickness (**r**); Tissue mineral density(**s**). Additionally, we also measured femur bone length (**t**). Results are shown as mean values ± SEM. The following statistical symbols were used to denote as follows. AAV8-CNP group vs. WT group, *** *p* ≤ 0.001; AAV8-CNP group vs. untreated MPS IVA group, ### *p* ≤ 0.001, ## *p* ≤ 0.01, # *p* ≤ 0.05.

**Table 1 ijms-24-09890-t001:** Pathological scores in bones of mucopolysaccharidosis type IVA (MPS IVA) mice treated with AAV8-NPPC vector.

Bone	Structure	Finding	MPS IVA	WT	AAV8-NPPC
Tibia	Growth plate	Vacuolization	2.9 ± 0.1	0.0	2.8 ± 0.2
		Column structure	2.8 ± 0.2	0.0	2.3 ± 0.3
	Articular cartilage	Vacuolization	2.9 ± 0.1	0.0	2.6 ± 0.1 #
		Column structure	2.8 ± 0.2	0.0	2.6 ± 0.1
Femur	Growth plate	Vacuolization	2.8 ± 0.2	0.0	2.6 ± 0.2
		Column structure	2.8 ± 0.2	0.0	2.6 ± 0.2
	Articular cartilage	Vacuolization	2.9 ± 0.1	0.0	2.9 ± 0.2
		Column structure	2.9 ± 0.1	0.0	2.7 ± 0.4
Ligament		Vacuolization	3.0 ± 0.0	0.0	2.4 ± 0.3
Meniscus		Vacuolization	2.9 ± 0.1	0.0	2.6 ± 0.2

Twelve weeks post-injection of the AAV8 vector, pathological scores were evaluated in femurs, tibias, ligaments, and meniscus from MPS IVA mice. Levels of storage materials and degrees of disoriented columns were scored. “No storage or very slight” was 0 (−), “slight but obvious” was 1 (+), “moderate” was 2 (++), and “marked” was 3 (+++). Each pathological slide was assessed in a double-blind manner three times. *n* = 4–8. Data are presented as means ± SDs. # *p* < 0.05 versus untreated MPS IVA.

## Data Availability

All data are available upon request.

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
