# Peer review of "Bone Growth Induction in Mucopolysaccharidosis IVA Mouse"

_ijms, 2023, doi:10.3390/ijms24129890_

Round 1

Reviewer 1 Report

The study of Rintz et al. describes a novel gene therapy in mucopolysaccharidosis IV A mice. The results suggest the potential of CNP peptide used as a treatment in MPS IV A patients. The manuscript is clearly written. The introduction provides sufficient background, methods are adequately described. Results are clearly presented. I have only several minor suggestions:

line 73-75: I suggest ending the introduction with the aims and predictions of your study rather than a summary of your results.

In Figures 2 c and d: there should be Length in the y-axis (not Lenght)

Figure 3: it will be better to separate j) into a new figure. BTW why is in j) an arbitrary unit and not a real unit? In Fig 3 a-i is the scale available…

Table 1. The decimal separator should be a dot, not a comma

Figure 4a: orange colour for the WT group is not visible. I guess it is under the grey one. Maybe you can do the dashed line, where both colours will be visible. Moreover, it will be nice to keep the same colour for groups as in other charts (blue, grey and black).

Fig 6 a,b,c,k,l,m scale bar missing

Line 338: should be gait, not gate

Line 365: scan should be in the past tense

In several places, there is MPS IV A without space between MPS and IV (e.g. lines 123, 125, Fig 6a). This should be corrected.

I can not see supplementary material in the system. Did you upload it?

Last but not least, did you do some power analysis? The sample size is not huge. Your study has very serious implications for further research, so it should be given attention to this issue.

English quality is well.

Author Response

Comments and Suggestions for Authors

The study of Rintz et al. describes a novel gene therapy in mucopolysaccharidosis IV A mice. The results suggest the potential of CNP peptide used as a treatment in MPS IV A patients. The manuscript is clearly written. The introduction provides sufficient background; methods are adequately described. Results are clearly presented. I have only several minor suggestions:

Line 73-75: I suggest ending the introduction with the aims and predictions of your study rather than a summary of your results.

Thank you for your comment. We corrected this part.

"The use of CNP peptide as a potential treatment for MPS IVA has not been previously tested for this condition. Our proposed therapy utilizes an AAV vector as a one-time treatment to express CNP peptides. This novel approach is advantageous because it eliminates the need for frequent administration. A single treatment with CNP delivered through a viral vector expression cassette could be a promising therapeutic approach for treating progressive skeletal dysplasia in MPS IVA patients.."

In Figures 2 c and d: there should be the length in the y-axis.

Thank you for noticing. We have corrected it.

Figure 3: it will be better to separate j) into a new figure. BTW why is in j) an arbitrary unit and not a real unit? In Fig 3 a-i is the scale available…

Thank you for your kind suggestion.

Unfortunately, we cannot separate just j) in the new figure as it may be harder for the reader.

It is an arbitrary unit as it was calculated based on the ImageJ program that generated an area number (not a real number). To make it more visible, we made an arbitrary unit also based on our previous article [10.1016/j.omtm.2020.05.015]. The method is described as stated:

"To evaluate chondrocyte cell size (vacuolization) in the growth plates of the tibia, approximately 300 chondrocytes in the proliferative area were measured in each mouse by Image J software, and results were expressed as folds-change from the wild-type group."

The scale is available on each of the pictures. (a), (d), and (g) have 10x magnification and a scale of 200 µm. (b), (e), (h), (c), (f), and (i) have 40x magnification and 50 µm.

Table 1. The decimal separator should be a dot, not a comma

Thank you for your comment. We corrected that part.

Figure 4a: orange color for the WT group is not visible. I guess it is under the grey one. Maybe you can do the dashed line, where both colors will be visible. Moreover, it will be nice to keep the same color for groups as in other charts (blue, grey, and black).

Thank you for noticing. We have corrected it.

Fig 6 a,b,c,k,l,m scale bar missing

Thank you for your comment. This is a representation of a 3D image from the micro CT analysis. It does not have a scale bar.

Line 338: should be gait, not gate

Thank you for your comment. We corrected that part.

Line 365: scan should be in the past tense

Thank you for your comment. We corrected that part.

In several places, there is MPS IV A without space between MPS and IV (e.g., lines 123, 125, Fig 6a). This should be corrected.

Thank you for your comment. We corrected that part.

I can not see supplementary material in the system. Did you upload it?

We have uploaded supplementary materials to the system. It includes figures as an addition to the manuscript (Figures S1, S2, S3).

Last but not least, did you do some power analysis? The sample size is not huge. Your study has very serious implications for further research, so it should be given attention to this issue.

We added a limitation section at the end of the manuscript for your comment.

"While our study provides valuable insights into the potential therapeutic use of CNP for progressive skeletal dysplasia in MPS IVA, there are several limitations to consider. Firstly, our study used a small sample size, which could affect the statistical power and significance of the conclusions drawn. Further studies with larger sample sizes may be necessary to confirm the effectiveness of CNP delivered through an AAV vector expression cassette. It should also be noted that the animal model we used did not exhibit the typical phenotype of MPS IVA patients. While none of the existing mouse models fully replicate the phenotype of human MPS IVA, our study used a model closer to growth plate pathology. Lastly, our study only evaluated the short-term effects of CNP treatment and did not assess potential toxicity or long-term outcomes. Further toxicity and long-term effects analysis will be necessary before CNP is delivered through an AAV vector expression cassette, which can be considered a viable therapeutic approach for progressive skeletal dysplasia in MPS IVA patients."

Reviewer 2 Report

The Authors present data to support development of a gene therapy approach to the treatment of mucopolysaccharidosis IVA. The human gene NPPC gene was packaged within AAV8 and tested by IV delivery within a murine model of MPS IVA. The approach is interesting and may be a successful program. However, there are several features that need to be addressed regarding this work.

11)      The AAV8 treatment resulted in the generation of larger & longer mice. How do the Authors explain this result?

22)      Table 1: only one finding was significant and no findings were near wild-type levels. This needs to be noted by the Authors because lines 122-125 does not convey the total data set.

33)      The biodistribution level of the liver vs bone is ~300-fold higher. This is expected with the AAV8 capsid. Can the Authors provide references for previous data of AAV8 reaching bone tissue in murine models?

44)      Lines 163-165: Given the biodistribution level in the liver, please clearly state whether CNP was detected in this tissue. Only bone tissue is specifically mentioned as not demonstrating detectable CNP.

55)      The Authors need to provide data of either mRNA expression or protein detection of the delivered gene or CNP, respectively. This is a critical aspect of the study. A cell assay result coupled with in vivo data would suffice. 

66)      There is no mention about how the AAV8 vector was produced. Details are not required, but a brief description of the system/process for how it was produced is relevant. Proprietary information should be omitted.

77) Have the Authors considered using a murine version of human gene? This may be more effective with in vivo results.

Author Response

Comments on the Quality of English Language

English quality is well.

The Authors present data to support the development of a gene therapy approach to the treatment of mucopolysaccharidosis IVA. The human gene NPPC gene was packaged within AAV8 and tested by IV delivery within a murine model of MPS IVA. The approach is interesting and may be a successful program. However, there are several features that need to be addressed regarding this work.

11)      The AAV8 treatment resulted in the generation of larger & longer mice. How do the Authors explain this result?

Thank you for your comment. The overexpression of CNP by placing the gene into an AAV8 cassette resulted in bone ossification, which inducted bone growth in the treated mice, which means a high expression of the CNP = generation of larger & longer mice.

We improved the introduction part to make it more straightforward for the reader.

22)      Table 1: only one finding was significant, and no findings were near wild-type levels. This needs to be noted by the Authors because lines 122-125 do not convey the total data set.

Thank you for your comment. We improved that part.

“The pathological score of MPSIVA mice treated with AAV8-NPPC was significantly lower for vacuolization of chondrocytes in the articular cartilage area compared to those treated with MPSIVA, but not as low as the WT level (as shown in Table 1). For other parameters, we observed a decrease, but it was not statistically significant. “

33)      The biodistribution level of the liver vs. bone is ~300-fold higher. This is expected with the AAV8 capsid. Can the Authors provide references for previous data on AAV8 reaching bone tissue in murine models?

Thank you for your comment. Our intention was not to target bone specifically using the AAV8 vector, as it has a tropism mainly directed toward the liver. We purposely chose a liver-targeted vector to achieve more systemic expression of CNP rather than bone-specific expression. We expected CNP expression to be high, and using a more general vector like AAV9 could lead to higher side effects. Additionally, our previous results with an AAV8 vector with a bone-targeted tag showed improvement in bone, which we added to the discussion section.

“The results were also parallel as in our other mouse models (MKC and MTOL, discussed in 10.1016/j.omtm.2020.05.015), tibia growth plate chondrocytes were found to be larger in MPS IVA knockout mice than in WT (Figure 3). Furthermore, our analysis delved more deeply into the level of GAG accumulation in affected tissues. We were the first to show mono-sulfated KS accumulation in the bone of the mouse MPS IVA model, in addition to other tissues. The difference in the level of mono-KS in the liver and lung (the only tissues shown in 10.1016/j.omtm.2020.05.015) was similar. In untreated MPS IVA, liver accumulation was 200 times higher than in WT, while in the lung, it was around 100 times more. These results are similar to our previous mouse models [10.1016/j.omtm.2020.05.015]. In the case of bone, we observed a 30% increase in mono-KS accumulation compared to WT mice (Figure 5).”

44)      Lines 163-165: Given the biodistribution level in the liver, please clearly state whether CNP was detected in this tissue. Only bone tissue is specifically mentioned as not demonstrating detectable CNP.

Thank your comment. We corrected that part.

“We performed immunohistochemistry staining to investigate the presence of CNP peptide in the liver and bone growth plate. However, we did not detect CNP in the bone growth plate samples, possibly because the peptide was secreted from the cells. In the liver samples, immunohistochemistry staining showed no significant difference in the expression levels between the mouse groups (data not shown). These results could be associated with using antibodies that cross-react with the murine NPPC gene.”

55)      The Authors need to provide data of either mRNA expression or protein detection of the delivered gene or CNP, respectively. This is a critical aspect of the study. A cell assay result coupled with in vivo data would suffice.

Per your suggestion, we performed a western blot analysis for the liver and arm bone. In the case of the liver, we did detect expression of the CNP in the liver. Bone protein level was insufficient to perform protein expression; thus, we do not have more material to analyze bone. Figure S4 is added to the Supplementary materials.

Figure S3. CNP protein expression in the liver tissue. Relative levels of the proteins were measured using the Western-blotting procedure. Representative blots are shown, and data are quantitated by densitometry. The following statistical symbols were used to denote as follows. AAV8-CNP group vs. WT group, * p ≤ 0.05.

We added a section in the manuscript describing protein expression in the liver:  

“In addition, we performed a Western blot to quantify the expression of the CNP protein in the liver, as its distribution level was over 3-fold higher than in WT mice. We detected CNP expression in all groups, including MPS IVA, WT, and AAV8-NPPC (shown in Figure S3). The antibody used detected both human and mouse CNP; however, we observed the highest expression in AAV8-NPPC treated mice due to the expression of both human and endogenous mouse CNP.”

Moreover, we updated the method section for the western blotting analysis.

“4.11. Western Blot

Protein lysate was isolated from both the liver and arm bone, as described in section 4.4. After lysis and determination of protein concentration, the protein extracts were separated by electrophoresis. However, we did not obtain a sufficient protein concentration in the arm tissue for further analysis. Therefore, only the liver tissue was analyzed for protein expression. After electrophoresis, the proteins were transferred onto a nitrocellulose membrane. The membrane was blocked with 5% nonfat dry milk in TBST buffer and then incubated with primary antibodies overnight at 4 °C (anti-CNP antibody from MyBioSource, USA, and beta-actin from Sigma-Aldrich, USA). The membrane was subsequently incubated with secondary antibodies at room temperature for 1 hour, treated with a solution of substrates for HRP detection, and read with a machine. As CNP and beta-actin have similar protein weights, the membrane was stripped after performing analysis for the CNP protein, and beta-actin analysis was performed starting from the blocking step. The intensities of bands were analyzed using the QuantityOne software.”

66)      There is no mention of how the AAV8 vector was produced. Details are not required, but a brief description of the system/process for how it was produced is relevant. Proprietary information should be omitted.

Thank you for your suggestion. We added this description to the method section.

"The AAV vector was produced by VectorBuilder (xxx). The quality control included titer measurement, sterility testing for bacteria and fungi, and mycoplasma detection. Additionally, we routinely sample virus quality by SDS-PAGE analysis and endotoxin assay

 for ultra-purified AAV.

For the recombinant AAV manufacturing, the transfer plasmid carrying the gene of interest (GOI) was co-transfected with the proprietary Rep-cap plasmid and helper plasmid encoding adenovirus genes (E4, E2A, and VA) that mediate AAV replication into HEK293T packaging cells. After a short incubation period, viral particles were harvested from cell lysate or supernatant depending on serotype and concentrated by PEG precipitation. For ultra-purified AAV (in vivo grade), viral particles were further purified and concentrated by cesium chloride (CsCl) gradient ultracentrifugation. We used a qPCR-based approach to measure AAV titer."

77) Have the Authors considered using a murine version of human gene? This may be more effective with in vivo results.

Thank you for your question. Human and murine NPPC genes do have similar sequences. We aim to test vectors in mouse models that will be further tested in humans; thus, using the human sequence of the gene is better in case of protocol optimization. Additionally, we are performing additional analysis on the expression of the human GALNS gene in the AAV cassette; thus, using the human NPPC gene and the human GALNS gene made more sense.